# A palisade-shaped membrane reservoir is required for rapid ring cell inflation in *Drechslerella dactyloides*

Yue Chen [1], Jia Liu [1], Seogchan Kang [2], Dongsheng Wei[1], Yani Fan [3,4] ✉, Meichun Xiang [3,4] ✉ & Xingzhong Liu [1,3] ✉

Fusion of individual vesicles carrying membrane-building materials with the plasma membrane (PM) enables gradual cell expansion and shape change. Constricting ring (CR) cells of carnivorous fungi triple in size within 0.1-1 s to capture passing nematodes. Here, we investigated how a carnivorous fungus, *Drechslerella dactyloides*, executes rapid and irreversible PM expansion during CR inflation. During CR maturation, vesicles carrying membrane-building materials accumulate and fuse, forming a structure named the Palisade-shaped Membrane-building Structure (PMS) around the rumen side of ring cells. After CR inflation, the PMS disappears, with partially inflated cells displaying wavy PM and fully inflated cells exhibiting smooth PM, suggesting that the PMS serves as the reservoir for membrane-building materials to enable rapid and extensive PM expansion. The DdSnc1, a v-SNARE protein, accumulates at the inner side of ring cells and is necessary for PMS formation and CR inflation. This study elucidates the unique cellular mechanisms underpinning rapid CR inflation.

Growth and development require genetically programmed cell division and various changes in cell shape and size. Intracellular vesicles carrying membrane-building materials fuse with the plasma membrane (PM) to expand the PM accompanying such changes[1,2]. Studies on the cells exhibiting polarized growth, such as animal neuronal cells, plant root hair and pollen tube, and fungal hyphae, have provided much of the available knowledge about how this process operates at the molecular and cellular levels. The formation of dendrites and axons and subsequent maintenance of the vastly expanded surface are achieved by directing and fusing vesicles carrying membrane-building materials to specific PM locations[3]. Similarly, such vesicles accumulate and fuse with the PM to drive the tip growth of plant root hairs and pollen tubes[4,5]. The apical growth of filamentous fungal hyphae requires the movement of secretory vesicles towards Spitzenkörper, a

multicomponent subcellular structure directing hyphal growth and morphogenesis, and subsequent fusion with the PM[6,7].

Membrane-vesicle fusion involves multiple protein families[8,9]. The exocyst complex is an evolutionarily conserved octameric protein complex that allows vesicles to tether to the PM prior to fusion[10,11], and Exo70, a component of this complex, is involved in axon growth[12,13]. Subsequent fusion is directed by two types of SNARE (soluble N-ethylmaleimide-sensitive factor attachment protein receptor) proteins: vesicle or v-SNARE associated with the vesicle membrane and target or t-SNARE localized at the PM[14]. These SNARE proteins form a four-helix bundle and pull the two membranes in a close proximity[15,16]. The SNARE proteins belong to a superfamily of small proteins, with 25, 54, and 36 members present in *Saccharomyces cerevisiae*, *Arabidopsis thaliana*, and humans, respectively[15,17]. Many are localized in distinct

[1]State Key Laboratory of Medicinal Chemical Biology, Key Laboratory of Molecular Microbiology and Technology of the Ministry of Education, Department of Microbiology, Frontiers Science Center for Cell Responses, College of Life Science, Nankai University, Tianjin 300071, China. [2]Department of Plant Pathology & Environmental Microbiology, The Pennsylvania State University, University Park, PA 16802, USA. [3]State Key Laboratory of Mycology, Institute of Microbiology, Chinese Academy of Sciences, Beijing 100101, China. [4]University of Chinese Academy of Sciences, Beijing 100049, China. ✉e-mail: hnfyn1234@163.com; xiangmc@im.ac.cn; liuxz@nankai.edu.cn

cellular compartments and are specialized for a specific fusion process (19). However, some likely participate in multiple fusion processes. The Snc proteins encoded by filamentous fungi are localized in hyphal tips and subapical compartments and regulate exocytosis in *Aspergillus oryzae* and *Trichoderma reesei*[18–20].

In contrast to gradual PM expansion associated with most cells, some cells rapidly increase cell volume and PM in response to mechanical and hypoosmotic stimuli by quickly mobilizing membrane-building materials stored in preexisting reserves, such as buds, wrinkles, folds, and invaginations[21–24]. However, these membrane reservoirs can only support limited PM expansion and cannot supply enough materials to enable extensive cell swelling. Some plants undergo rapid macroscopic structural changes accompanied by cell shape and volume changes in response to external stimuli[25,26]. Well-studied examples include the stomatal movement and the fast movement of trap leaves in Venus flytrap, a carnivorous plant[27–29]. The physical mechanism underpinning the function of such specialized organs and the nature and mode of action of signaling molecules involved have been extensively characterized[25,26]. However, the mechanism underlying rapid shape and volume changes at the single-cell level has not received adequate attention.

The constricting ring (CR) is a sophisticated trapping device evolved by some carnivorous fungi to capture and consume nematodes[30,31]. It consists of three ring cells and two stalk cells, with the ring cells capable of inflating within 0.1–1 s to capture a nematode passing through the ring lumen[31–33] (Supplementary Movie 1). After inflation, the ring cell volume and surface area increase by 300% and 50%, respectively[33,34]. Simple cell composition and the availability of fungal gene manipulation tools make the CR an excellent model for studying the mechanism underpinning the rapid and extensive volume and surface change at the single cell level. Because the maximum possible stretching of the PM without incorporating new membrane-building materials is limited to only ~2%[35], such a drastic PM change must require a large quantity of new materials. A structure described as "labyrinthine networks" in uninflated ring cells was proposed to serve as membrane reservoirs[36]. We investigated the formation and function of this structure in *Drechslerella dactyloides*, a carnivorous fungus that differentiates CRs from mycelia within a few hours of contact with nematodes[30,37]. Newly formed CRs could not immediately inflate and required several hours of maturation to be capable of inflation. Detailed microscopic observations of CRs and their subcellular components during CR maturation and inflation revealed that a membrane structure, named the Palisade-shaped Membrane-building Structure (PMS), forms on the inner side of the ring cells during maturation and supplies membrane-building materials. The involvement of cytoskeleton components in PMS formation was evaluated using specific chemical inhibitors. We also investigated the role of two genes encoding v-SNARE proteins (*DdSnc1* and *DdSnc2*) and the *Exo70* homolog *DdExo70* in PMS formation and ring cell inflation via targeted mutagenesis and fluorescent protein tagging.

## Results

### Morphogenesis and development of CR

CR morphogenesis from formation to inflation and accompanied subcellular changes were microscopically examined. CRs differentiate from hyphae and develop two basal stalk cells (S1 and S2) and three distal ring cells (R1, R2, and R3) (Fig. 1a, b). The R3 cell fuses with the S2 and R1 cells to complete CR formation (Fig. 1a, c and Supplementary Movie 2). A branched hypha completes CR formation in ~4 h. Approximately 5 h after CR formation, it could be stimulated to inflate by applying hot water (55 °C) (Fig. 1a). The ring cells could rapidly (0.1–1 s) inflate inward to capture a nematode passing through the lumen of CR, rupturing the outer wall (OW), stretching the inner wall (IW), and drastically increasing cell surface area and volume (Fig. 1d–f and Supplementary Movies 1 and 3). The large vacuoles and PM

remained intact, with the nuclei squeezed to the cell periphery by enlarged vacuoles (Fig. 1g, h), suggesting that the ring cells remain viable after inflation.

CR formation can be induced by vermiform nematodes, and ring cell inflation can be triggered by a passing nematode or hot water (55 °C)[31]. Numerous CRs were fully formed 16 h after introducing *Caenorhabditis elegans*. However, hot water treatment or nematodes could not inflate these newly formed CRs immediately after their formation (Fig. 1i, j and Supplementary Fig. 1). New CRs continuously form between 16 and 36 h, and some inflated CRs were observed after 36 h (Fig. 1i, j). In the presence of *C. elegans*, the percentage of inflated CRs continuously increased, with almost all CRs inflated after 72 h (Fig. 1i, j and Supplementary Fig. 1). Few nematodes were captured within 16 h after introducing *C. elegans* (Fig. 1k, l and Supplementary Movie 4). However, most nematodes were caught after 36 h (Fig. 1k, m and Supplementary Movie 1). Microscopic observations of individual CRs over time indicated that newly formed CRs require at least 5 h to become inflatable (Fig. 1a). However, there was no morphological difference between immature and mature CRs (Fig. 1a), suggesting that newly formed CRs should undergo some subcellular changes to become inflatable. We classified the development of CR into three stages (growth, maturation, and inflation) and divided CRs into three types, immature (morphologically complete but incapable of inflation), mature (competent to inflate), and inflated (Fig. 1a), to help quantify the progression of CR development and inflation under different conditions.

### A subcellular structure forms via multi-vesicle fusion during CR maturation

Since the PM resists stretching, a large quantity of new membrane-building materials should be deposited to enable CR inflation without losing cell integrity. Considering that de novo synthesis during CR inflation is unlikely, we hypothesized that ring cells in newly formed CRs would produce and store membrane-building materials in a quickly deployable form. We analyzed ultrastructural features and changes of the ring cells during CR maturation and inflation as outlined in Fig. 2a. The PM of the cells of newly formed CRs looked smooth and straight (Fig. 2b and Supplementary Fig. 2a). During maturation, many vesicles, presumably carrying membrane-building materials, accumulate near the PM of the inner side of individual ring cells (Fig. 2c–e and Supplementary Fig. 2b, c). These vesicles were similar in size, shape, and electron density, with some being fused with other vesicles (Fig. 2f, g). In immature CRs, individual vesicles or fused vesicles did not connect with the PM (Fig. 2f, g). In mature CRs, strings of three or more fused vesicles form a larger structure, named the Palisade-shaped Membrane-building Structure (PMS), oriented in a specific direction (Fig. 2h–k), with some of the strings conjugated with the PM (Fig. 2h, j and Supplementary Fig. 2d). Compared to immature CRs, the inner side of the ring cells of mature CRs stained with fluorescent membrane dye FM4-64 was distinct from the outer side, with more membrane being present on the inner side, supporting that PMS formation brought extra membrane-building materials (Supplementary Fig. 3a). Longitudinal and transverse sections of mature CRs showed that the PMS was present only on the inner side of ring cells where the cell surface expands during nematode trapping (Fig. 2h–j). In addition, the cell wall of mature CRs was asymmetrical, with the wall being significantly thicker on the inner side (Fig. 2i).

### The PMS is involved in the extensive PM expansion associated with ring cell inflation

Application of 1% NaCl solution at 55 °C to mature CRs partially inflated ring cells (Supplementary Fig. 3b). The PMS disappeared in partially inflated ring cells, with their PM being wavy and curved (Fig. 3a and Supplementary Fig. 3c). The PMS also disappeared in fully inflated ring cells, but their PM was smooth and uniform (Fig. 3b, c and

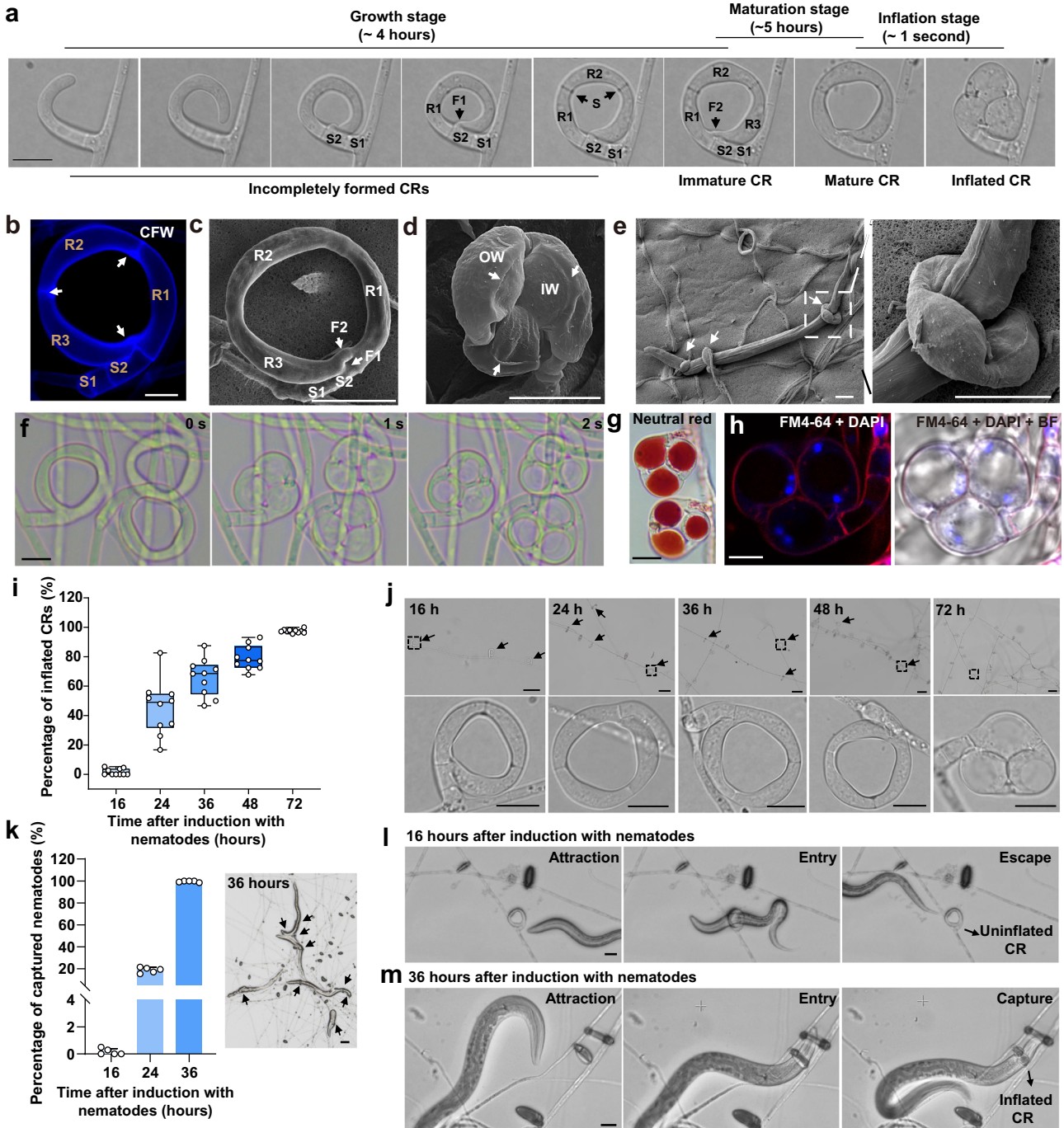

**Fig. 1 | Cytological and morphological characteristics and changes during CR formation, maturation, and inflation. a** Time-lapse images documenting CR formation, maturation, and inflation. S, septum; R1, R2, and R3, ring cells; S1 and S2, stalk cells; F1, first fusion between R3 and S2; F2, second fusion between R3 and R1. Scale bar = 10 μm. **b** Ring and stalk cells stained with CFW. White arrows denote septa. Scale bar = 6 μm. **c**−**e** Scanning electron micrographs of an uninflated CR (**c**), an inflated CR (**d**), and a nematode captured by three CRs (noted by white arrows) (**e**). OW, outer cell wall exhibiting ruptured areas (denoted by white arrows in **d**). IW, inner cell wall. Scale bars = 10 μm. **f** The inflation process of CRs (see Supplementary Movie 3). Scale bar = 10 μm. **g** Vacuoles stained using neutral red. Scale bar = 10 μm. **h** The PM and nuclei of ring cells stained using FM4−64 (red) and DAPI (blue), respectively. BF, bright field. Scale bar = 6 μm. **i** Box plot showing the percentages of inflated CRs after applying hot water (*n* = 10 microscopic fields of view

from 2 biological replicates). The middle line in the boxplot displays the median, edges represent the upper and lower quartiles, and whiskers indicate minimum and maximum values. **j** Uninflated CRs after applying hot water. Scale bars = 50 μm (upper), black arrows indicate uninflated CRs. The lower panel (scale bars = 10 μm) shows magnified views noted by a dotted black box. **k** Bar chart showing the percentages of nematodes captured after its introduction (*n* = 5 biological replicates). Data are means ± SD. Light micrograph showing trapped nematodes after 36 h. Black arrows denote inflated CRs. Scale bar = 50 μm. **l** Time-lapse images of a nematode entering and escaping from an immature CR (see Supplementary Movie 4). Scale bar = 10 μm. **m** Time-lapse images of a nematode trapped by a mature CR (see Supplementary Movie 1). Scale bar = 10 μm. Source data are provided as a Source Data file.

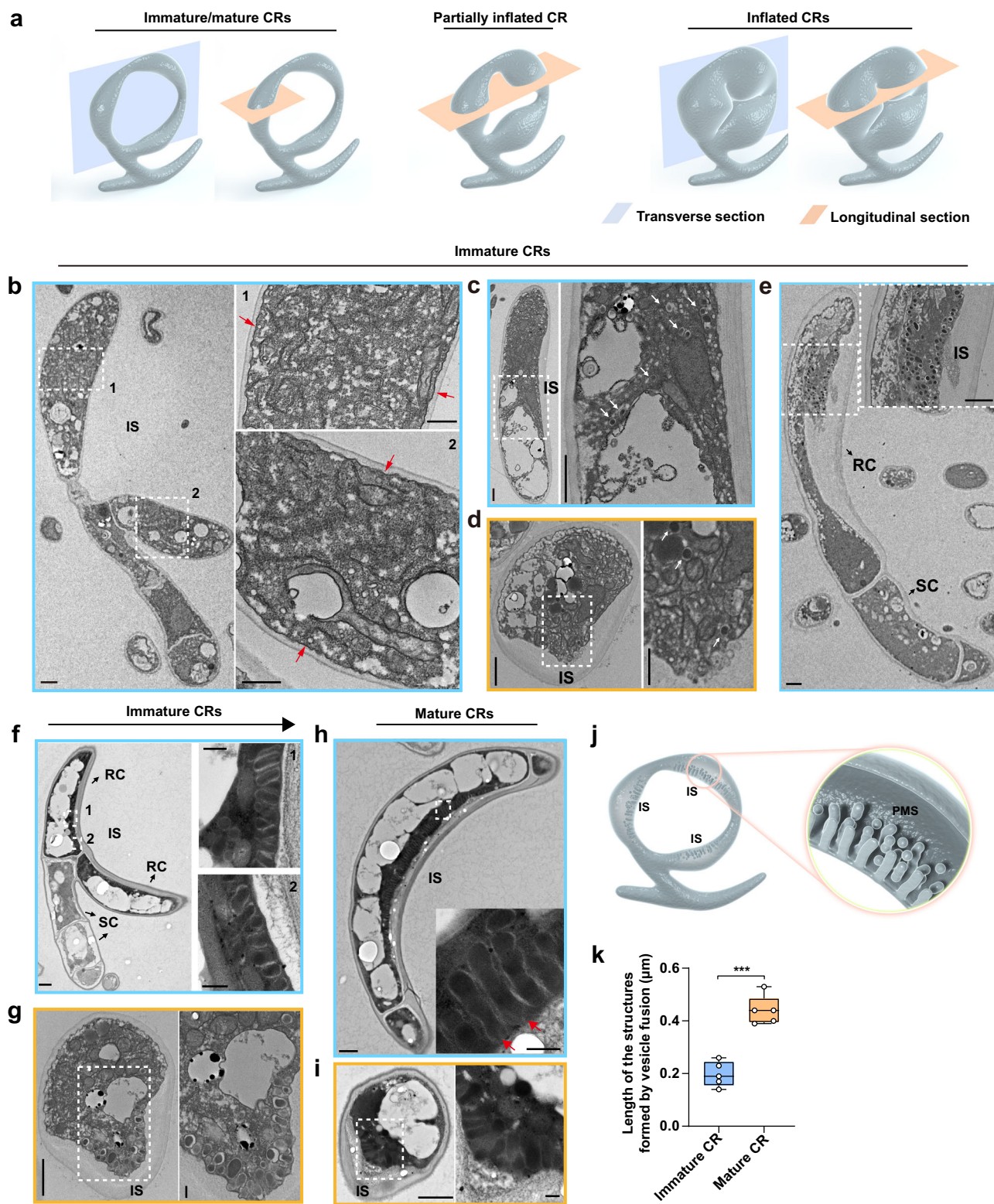

Supplementary Fig. 3a). After ring cell inflation, the outer layer of the thickened cell wall was ruptured, while the inner layer was intact but stretched (Figs. 1d and 3c). In inflated ring cells, enlarged vacuoles squeezed the cytoplasm tightly against the PM, and the PM looked adhered to the cell wall (Fig. 3b, c). These structural changes suggested that the PMS functions as the membrane reservoir and enables rapid and extensive PM expansion by incorporating its membrane cargo into the PM during ring cell inflation.

Brefeldin A (BFA), an inhibitor of vesicular trafficking[38], generally affect the initial secretion pathway (ER-Golgi transportation)[39]. After treating immature and mature CRs with BFA for 24 h, the percentage of inflated CRs in response to hot water treatment was measured to determine whether vesicular trafficking is critical for CR maturation/inflation. The BFA treatment did not affect the inflation of mature CRs (Fig. 3d). However, the percentage of inflated CRs was significantly decreased when the treatment occurred before CRs maturation

**Fig. 2 | A subcellular structure forms via multi-vesicle fusion during CR maturation.** The blue and yellow borders denote the transverse and longitudinal sections, respectively. Magnified views of the areas denoted by the white dotted box in (**b**–**i**) are shown on the right side. **a** Schematic diagrams illustrating how the transverse and longitudinal sections of immature, mature, partially inflated, and inflated CRs were selected for imaging. **b** An immature CR. Red arrows denote PM. Scale bars (left panel = 1 μm; right panel = 0.5 μm). **c**–**e** Immature CRs showing vesicles (noted by white arrows) transported to the inner side (IS) of ring cells. RC, ring cell; SC, stalk cell. Scale bars (left panel = 1 μm; right panel = 0.5 μm). **f**, **g** Immature CRs containing fused vesicles before they contact the PM of the inner side (IS) of ring cells. Scale bars (left panel = 1 μm; right panel = 0.2 μm). **h**, **i** Mature

CRs with the PMS on the inner side (IS) of ring cells. Some strings of fused vesicles were connected to the PM (red arrows). Scale bars (left panel = 1 μm; right panel = 0.2 μm). The images shown in (**b**–**i**) were chosen among those collected from at least 5 different CRs. **j** Schematic diagram illustrating the PMS on the inner side (IS) of ring cells. **k** Box plot showing the lengths of strings of fused vesicles in immature and mature CRs (n = 5 CRs per groups). In each ring cell, the lengths of individual strings were measured at three randomly selected locations. Two-tailed t-test. ***P = 0.000061. The middle line in the boxplot displays the median, edges represent the upper and lower quartiles, and whiskers indicate minimum and maximum values. Data shown are representatives from experiments performed in triplicate. Source data are provided as a Source Data file.

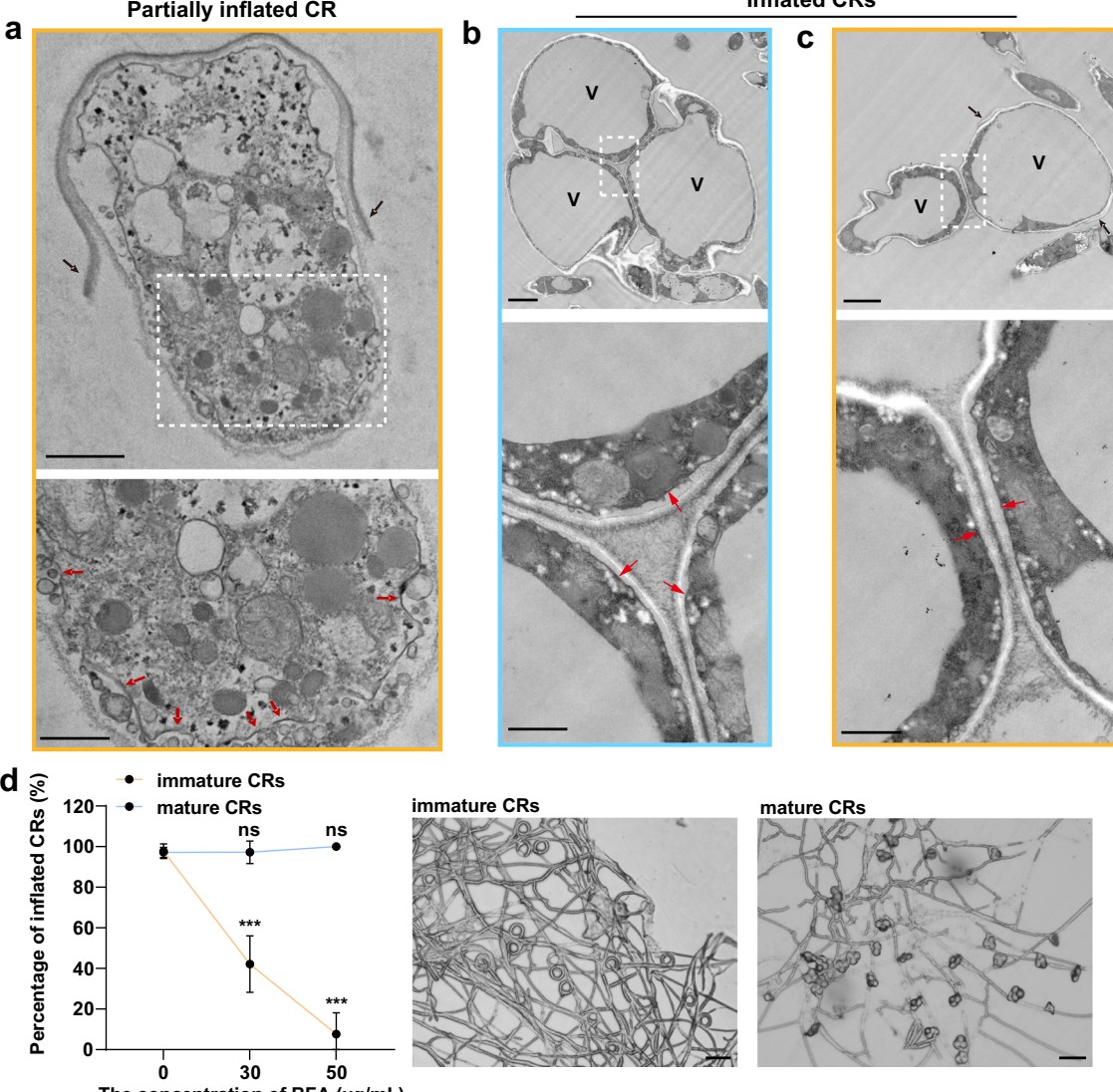

**Fig. 3 | Involvement of the PMS in the extensive PM expansion associated with ring cell inflation.** The blue and yellow borders denote the transverse and longitudinal sections, respectively. Magnified views of the areas denoted by the white dotted box in (**a**–**c**) are shown at the bottom. **a** The wavy and curved PM (red arrow) and the ruptured outer cell wall (black arrow) of a partially inflated CR. Scale bars (upper panel = 1 μm; lower panel = 0.5 μm). **b**, **c** The PM (red arrow) and the ruptured outer cell wall (black arrow) of inflated CRs. V, vacuole. Scale bars (upper panel = 2 μm; lower panel = 0.5 μm). The images shown in (**a**–**c**) were chosen among

those collected from at least 5 different CRs. **d** Percentages of inflated CRs after exposing immature CRs and mature CRs to 30 μg/mL and 50 μg/mL BFA for 24 h (n = 5, 4 biological replicates for immature, mature CRs, respectively). Two-tailed t-test, mean ± SD. ***P = 2.49 × 10⁻⁵ (30 μg/mL), 9.14 × 10⁻⁸ (50 μg/mL); ns not significant (P = 0.98, 0.06 for 30, 50 μg/mL, respectively). Light micrographs of mycelia and CRs after applying 50 μg/mL BFA. Scale bars = 50 μm. Source data are provided as a Source Data file. Data shown are representatives from experiments performed in triplicate.

(Fig. 3d), supporting the involvement of vesicle trafficking in CR maturation/inflation. The cytoskeleton is involved in vesicular trafficking[6,7]. The percentage of inflated CRs after treating immature CRs with benomyl, a microtubule polymerization inhibitor, significantly decreased (Supplementary Fig. 4a, b). However, latrunculin B, an actin polymerization inhibitor, did not impair the inflation capability of CRs (Supplementary Fig. 4c, d). These results suggest that the long-range microtubule-mediated vesicle trafficking, but not the short-range vesicular transport, is critical for PMS formation and, consequently, CR inflation.

### Characterization of the role of three *D. dactyloides* genes associated with vesicle fusion

To investigate the genetic mechanism underlying PMS formation and PM expansion, we disrupted three *D. dactyloides* genes homologous to those known to mediate vesicle fusion, including *DdExo70*, a homolog of the exocyst component-encoding gene *Exo70* (Supplementary Fig. 5), and *DdSnc1* and *DdSnc2*, homologs of the yeast v-SNARE-encoding genes *Snc1* and *Snc2* and human homologs *Vamp4* and *Vamp7*[40] (Supplementary Fig. 6a). Among the 16 putative SNARE-encoding genes previously identified in *D. dactyloides*[41], we chose the two *Snc* genes because their products are localized in secretory vesicles and involved in vesicle fusion in yeast[40,42]. The mutants created via targeted gene disruption were confirmed via Southern analysis and the absence of their transcripts (Supplementary Fig. 7). Deletion of *DdExo70* reduced colony growth and abolished conidiation (Supplementary Fig. 8a–c). The Δ*DdExo70* mutant rarely formed CRs (Supplementary Fig. 8d, e), and those formed were often deformed and could not inflate even 72 h after introducing nematodes (Supplementary Fig. 8f, g). These defects could be complemented when a copy of *DdExo70* was introduced into the mutant.

Colony growth of the Δ*DdSnc1* and Δ*DdSnc2* mutants was slightly defective (Fig. 4a, b), and their conidiation was not affected (Fig. 4c). However, when both genes were deleted, colony growth and conidiation were severely impaired (Fig. 4a–c), suggesting their functional redundancy. CR formation was significantly reduced in both Δ*DdSnc1* and Δ*DdSnc2* (Fig. 4d, e). Most CRs formed by Δ*DdSnc1* failed to inflate in response to 55 °C water treatment at 72 h after inoculating nematodes, whereas ~95% CRs formed by the wild-type (WT) strain inflated (Fig. 4f, h). In some cases, CRs with one or more partially inflated ring cells were observed in Δ*DdSnc1* (Supplementary Fig. 9). In *DdSnc1*^C, a complemented strain of Δ*DdSnc1*, CR inflation was fully restored (Fig. 4f, h). However, deletion of *DdSnc2* did not impair CR inflation, with the inflation rates for Δ*DdSnc2* and WT reaching ~97% and ~98%, respectively, at 72 h after inoculating nematodes (Fig. 4g, h). Unlike Δ*DdExo70*, no deformed CRs were observed in Δ*DdSnc1* and Δ*DdSnc2* (Fig. 4h). While the WT and *DdSnc1*^C strains captured all nematodes 36 h after induction, no nematodes were captured by Δ*DdSnc1* (Fig. 4i–l and Supplementary Movies 5–7). Although Δ*DdSnc1* CRs could attract nematodes to pass through the lumen, they failed to inflate to capture them (Fig. 4k and Supplementary Movie 6). The Δ*DdSnc1*Δ*DdSnc2* mutant failed to form CRs 36 h after introducing nematodes (Fig. 4d, e) but formed a few 6 days after introduction (Supplementary Fig. 10a, b). However, these CRs could not inflate in response to nematodes or hot water (Supplementary Fig. 10b–d). These results suggested that although *DdSnc1* and *DdSnc2* are redundant in supporting vegetative growth, only *DdSnc1* is required for CR inflation.

### DdSnc1 is required for PMS formation

DdSnc1, which consists of 119 amino acids, contains highly conserved residues in the signature C-terminal SNARE domain and a predicted transmembrane helix (Supplementary Fig. 6b). We used the DdSnc1 protein fused to RFP to track its location during CR morphogenesis. The fusion protein was localized at hyphal tips (Fig. 5a) and exhibited punctate distribution in the ring cells of incompletely formed CRs (Fig. 5b). Notably, DdSnc1 mainly accumulated at the inner rim of the ring cells during CR maturation (Fig. 5c). In mature CRs, it was enriched in the vacuoles in addition to the inner side of the ring cells (Fig. 5d). The DdSnc2 protein fused to GFP was also localized at hyphal tips (Fig. 5e), exhibited punctate distribution in the ring cells of incompletely formed CRs (Fig. 5f), but was enriched only in the vacuoles of the ring cells of immature and mature CRs (Fig. 5g, h). In the transformants expressing RFP only, the RFP signal was diffused throughout the cytoplasm of hyphal tips, incompletely formed CRs, and immature CRs (Supplementary Fig. 11a–c) and accumulated in the vacuoles of mature CRs (Supplementary Fig. 11d). These results suggested that both DdSnc1 and DdSnc2 are involved in tip growth. However, only DdSnc1 participates in PMS formation during CR maturation. Transcription of *DdSnc2* was upregulated in Δ*DdSnc1* cultured on WA, and that of *DdSnc1* was upregulated in Δ*DdSnc2* (Fig. 5i, j), further suggesting that *DdSnc1* and *DdSnc2* perform some redundant functions in vegetative growth.

CRs of the WT, Δ*DdSnc1*, and *DdSnc1*^C strains were analyzed to determine if DdSnc1 is involved in the membrane fusion process needed for PMS formation. The diameters of their CRs looked comparable (Fig. 6a), and all of them contain three ring cells and two stalk cells (Fig. 6b). Just like the WT and *DdSnc1*^C strains, two successive cell fusion events associated with CR formation (Fig. 1a, c) occurred in Δ*DdSnc1* (Fig. 6c), arguing against the possibility that the inability of the Δ*DdSnc1* ring cells to inflate was due to defective CR formation[37]. TEM analysis showed the absence of PMS on the inner side of the Δ*DdSnc1* ring cells and the restoration of PMS formation in *DdSnc1*^C (Fig. 6d), supporting the hypothesis. The cell wall on the inner side of the ring cells was thickened in Δ*DdSnc1* like the WT and *DdSnc1*^C strains (Fig. 6d). The inner cell wall of Δ*DdSnc1*'s partially inflated ring cells appeared indistinguishable from that of inflated WT ring cells based on staining with galanthus nivalis (GNL), concanavalin A (ConA), and calcofluor white (CFW) (Supplementary Fig. 12), suggesting that the loss of *DdSnc1* did not seem to affect cell wall composition and remodeling during CR formation.

## Discussion

Cells undergo genetically programmed shape/size changes to perform specific functions or orderly growth and differentiation. PM remodeling associated with such changes is typically driven by two strategies - supplying new membrane-building materials via secretory vesicle fusion and membrane unfolding without adding new materials[3,4,6]. Here, we show that *D. dactyloides* forms PMS, a large membrane reservoir, to enable rapid and extensive PM expansion accompanying CR inflation. The disappearance of PMS in both partially and fully inflated ring cells (Fig. 3a–c) suggests that its membrane-building materials were incorporated into the PM during CR inflation. The PM of partially inflated ring cells appeared wavy and curved because the cell surface was not fully expanded. The PMS may not directly regulate CR inflation but is required to keep ring cells from losing integrity and viability after the rapid and drastic inflation, enabling *D. dactyloides* to infect and digest trapped nematodes[31]. The insights from studying the mechanism of PMS formation will help future studies on the mechanism of PM expansion, remodeling, and repair and how these processes have evolved.

The PMS performs a function that is not required in most cells and organisms. The extent of PM expansion (up to a 50% increase of the ring cell surface area) in a confined space (inner side of ring cells) is unique. However, it also exhibits some features observed in previously identified mechanisms of PM remodeling. Since the vesicles forming PMS are coated with a membrane, the PMS resembles buds, wrinkles, folds, and invaginations that support membrane expansion in other types of cells. Many linked vesicles (3–4 vesicles) cluster in a specific orientation on the inner side of ring cells and then fuse with the PM

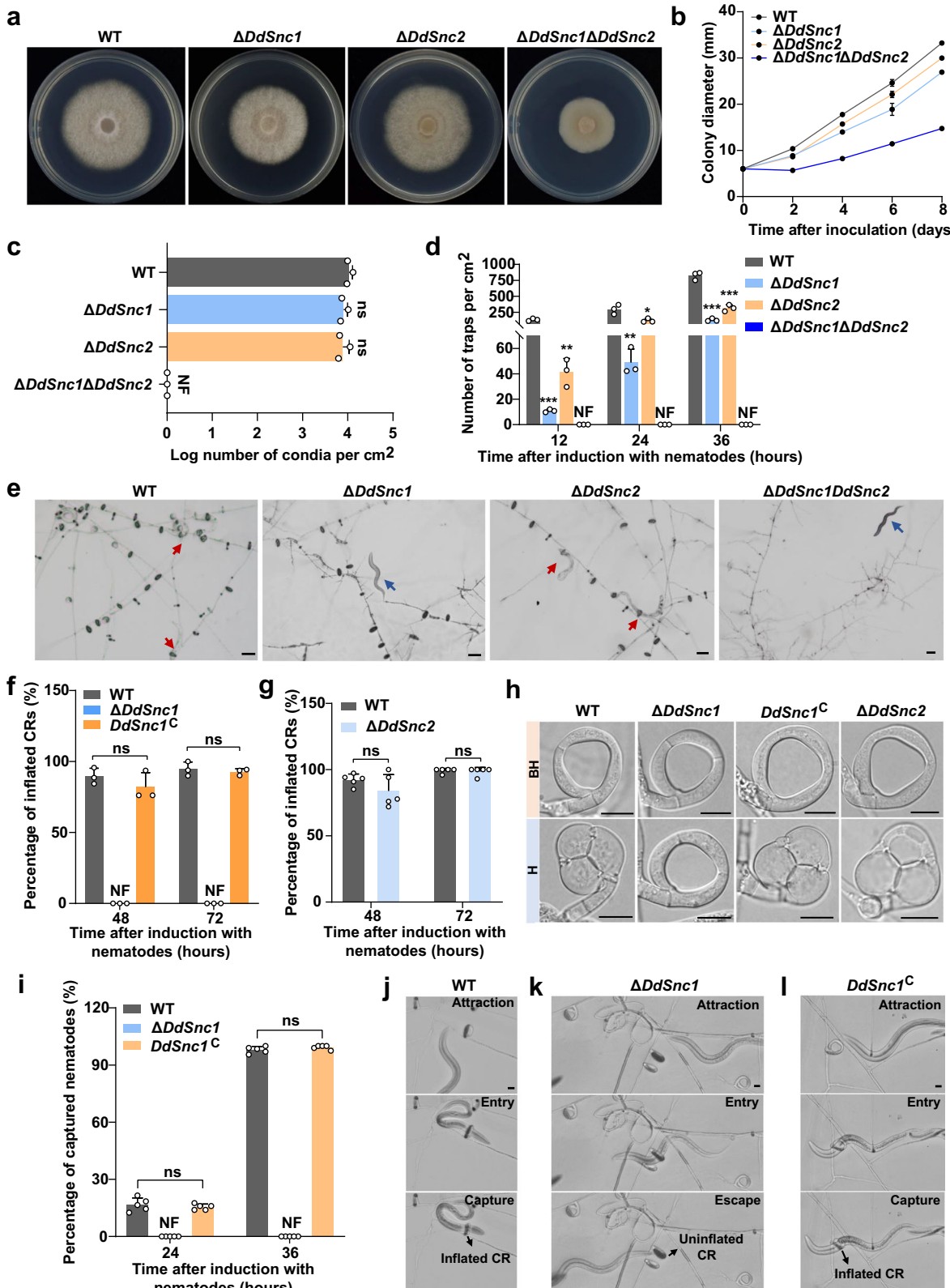

(Fig. 2h and Supplementary Fig. 2d). This process is reminiscent of compound exocytosis, a specialized form of secretion in which vesicles undergo fusion with each other and the PM. Compound exocytosis has been observed in various cell types and accounts for up to 30% PM increase during exocytosis in mast cells[43,44]. Homotypic vesicle-vesicle fusion has been shown to create a patch vesicle for resealing disrupted PM, a commonly occurring event in many cells[45].

Several SNARE proteins have been shown to be involved in coordinating vesicle-vesicle and vesicle-PM fusions. In pancreatic acinar cells, the v-SNARE protein VAMP8 is present on secretory vesicles, and loss of VAMP8 causes defects in secretion and reduces vesicle-to-vesicle fusion[46,47]. In *T. reesei*, the Snc1 protein is localized in the cluster of vesicles near growing hyphal tips[19]. The *A. oryzae* Snc1 protein is involved in tip growth and is localized at the septum, where it regulates

**Fig. 4 | Phenotypes caused by the loss of *DdSnc1*, *DdSnc2*, or both. a–c** Colony morphology of the WT and three mutant strains after culturing eight days on PDA (**a**) and their growth rate (**b**) and conidiation (**c**). Two-tailed *t*-test. The means ± SD of three biological replicates are shown. ns, not significant (In (**c**) *P* = 0.14, 0.19 for Δ*DdSnc1*, Δ*DdSnc2*, respectively); NF not found. **d** Bar chart showing the numbers of CRs formed per cm$^2$ at 12, 24, and 36 h after introducing nematodes. Two-tailed *t*-test. The data shown are the means ± SD of three biological replicates. For Δ*DdSnc1*, *P* = 0.0004 (12 h), 0.005 (24 h), 0.00006 (36 h). For Δ*DdSnc2*, *P* = 0.0024 (12 h), 0.02 (24 h), 0.0004 (36 h). NF, not found. **e** Light micrographs showing CR formation and inflation 36 h after introducing nematodes. Red and blue arrows denote captured and free nematodes, respectively. Scale bars = 50 μm. **f, g** Bar chart showing the percentages of inflated CRs after applying 55 °C water 48 and 72 h after

introducing nematodes (In (**f**) n = 3 biological replicates; in (**g**) *n* = 5 biological replicates). Two-tailed *t*-test, mean ± SD. ns not significant (In (**f**) *P* = 0.32, 0.59 for 48 h, 72 h, respectively; in (**g**) *P* = 0.20, 0.73 for 48 h, 72 h, respectively). NF not found. **h** CR morphologies before (BH) and after (H) heat stimulation. Scale bars = 10 μm. **i** Bar chart showing the percentages of captured nematodes 24 and 36 h after its introduction (*n* = 5 biological replicates). Two-tailed *t*-test, mean ± SD. ns not significant (*P* = 0.50, 0.20 for 24 h, 36 h, respectively); NF not found. Source data are provided as a Source Data file. **j** A nematode trapped by a CR of WT (see Supplementary Movie 5). **k** A nematode entering and escaping a CR of Δ*DdSnc1* (see Supplementary Movie 6). **l** A nematode trapped by a CR of *DdSnc1*$^C$ (see Supplementary Movie 7). The scale bars for (**j–l**) are 10 μm. Data shown are representatives from duplicated experiments.

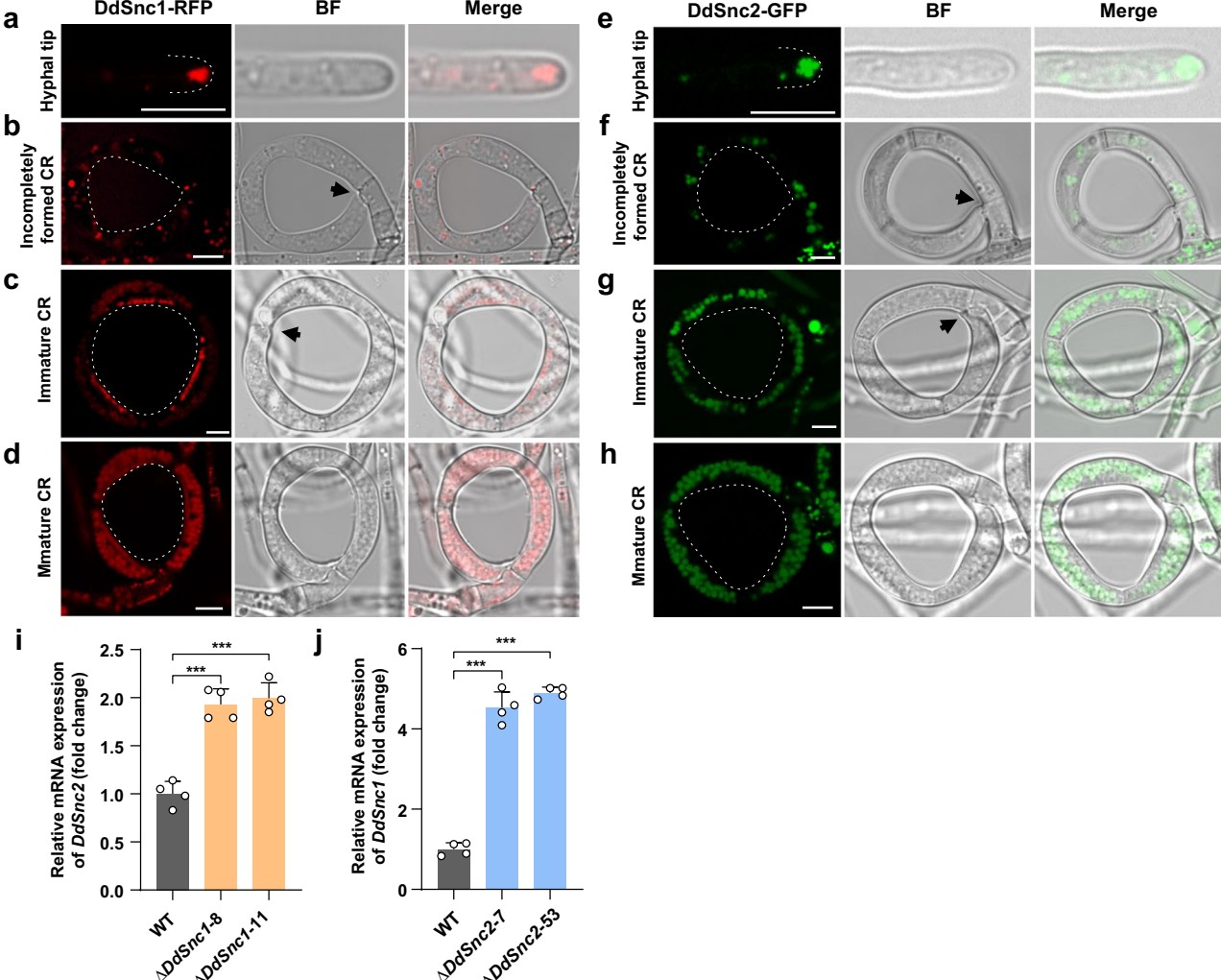

**Fig. 5 | Localization of DdSnc1 and DdSnc2 fused with RFP and GFP, respectively. a, e** Hyphal tips. The dotted white line denotes the hyphal tip. **b, f** Incompletely formed CRs (partially completed fusion was noted by black arrow). **c, g** Immature CRs with completed fusion (noted by black arrow). **d, h** Mature CRs. The dotted white circle in (**b–d, f–h**) denotes the inner side of CR. The scale bars for (**a–h**) are 6 μm. **i** Levels of *DdSnc2* transcripts in two Δ*DdSnc1*

mutants during vegetative growth (*n* = 4 independent experiments). Two-tailed *t*-test, mean ± SD. ***P* = 0.00011 (Δ*DdSnc1*–8), 7.12 × 10$^{-5}$ (Δ*DdSnc1*–11). **j** Levels of *DdSnc1* transcripts in two Δ*DdSnc2* mutants during vegetative growth (*n* = 4 independent experiments). Two-tailed *t*-test, mean ± SD. ***P* = 3.03 × 10$^{-6}$ (Δ*DdSnc2*–7), 2.93 × 10$^{-8}$ (Δ*DdSnc2*–53). Source data are provided as a Source Data file. The data shown here are representatives from duplicated experiments.

the fusion of secretory vesicles with the septal PM[18]. The v-SNARE protein DdSnc1 is homologous to animal Vamp and yeast Snc proteins. The Snc proteins are located on the vesicle membrane in yeast[40], and DdSnc1 was located at the inner side of ring cells where multi-vesicle fusion occurred (Fig. 5c). Δ*DdSnc1* failed to perform PMS formation and inflate ring cells (Fig. 6d), supporting the critical role of *DdSnc1* in

PMS formation and the importance of PMS in ring cell inflation. Colony growth and conidiation were only slightly reduced when either *DdSnc1* or *DdSnc2* was deleted. However, when both were deleted, growth and conidiation were severely reduced (Fig. 4a–c), suggesting that DdSnc1 and DdSnc2 perform redundant functions during vegetative growth. Both DdSnc1 and DdSnc2 were localized at hyphal tips (Fig. 5a, e), and

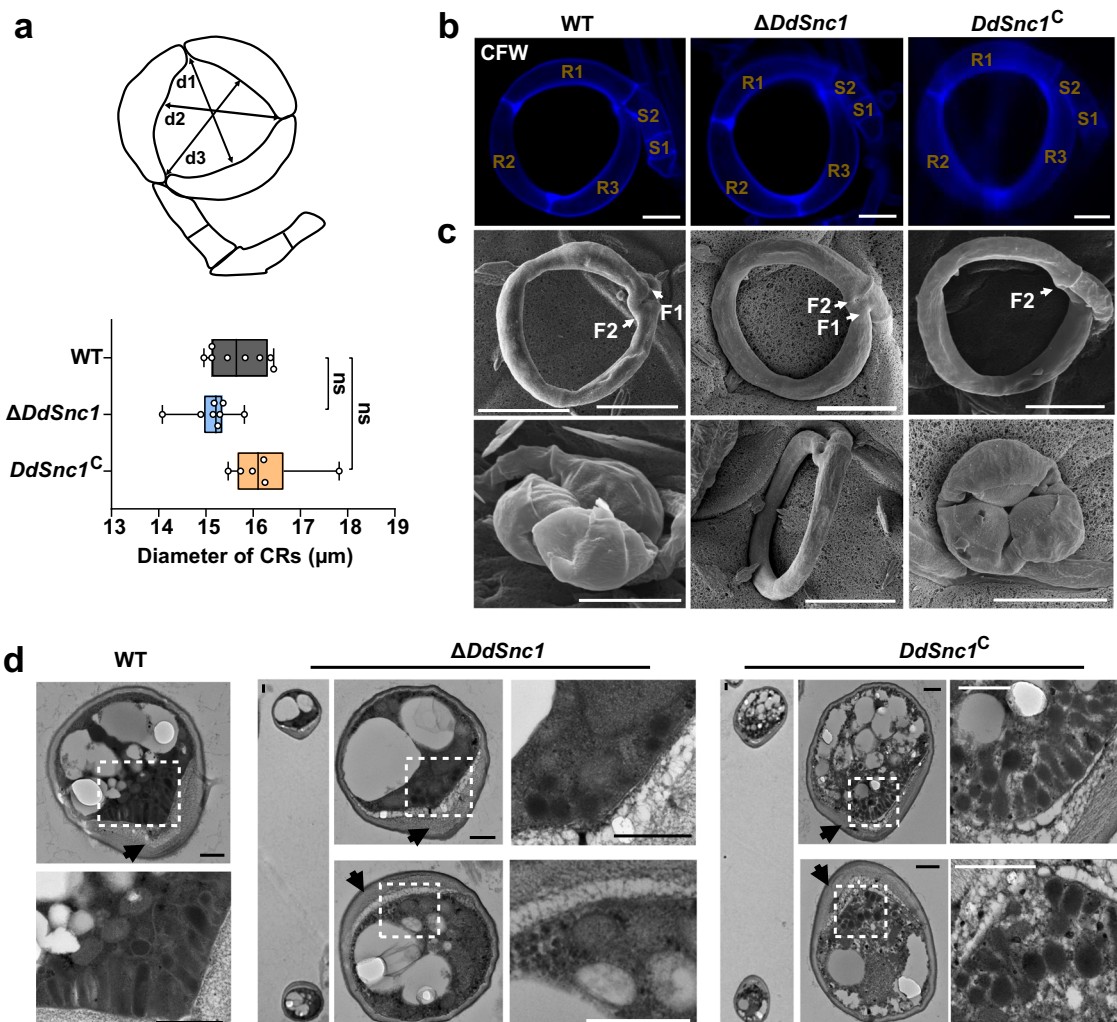

**Fig. 6 | Requirement of DdSnc1 for PMS formation. a** Box plot showing the diameter of CRs formed by the WT, ΔDdSnc1, and DdSnc1[C] (*n* = 8 CRs for WT and ΔDdSnc1, and *n* = 6 CRs for DdSnc1[C]). The diameter was measured at d1, d2, and d3 noted in the schematic diagram. Two-tailed *t*-test. ns not significant (*P* = 0.06, 0.16 for ΔDdSnc1, DdSnc1[C] respectively). The middle line in the boxplot displays the median, edges represent the upper and lower quartiles, and whiskers indicate minimum and maximum values. Source data are provided as a Source Data file. Data shown are representatives from experiments performed in triplicate. **b** The ring cells (R1, R2, and R3) and stalk cells (S1 and S2) of CRs formed by WT, ΔDdSnc1, and DdSnc1[C] were stained with CFW. Scale bars = 6 μm. **c** Scanning electron micrographs of CRs before (upper panel) and after (lower panel) heat stimulation. F1 and F2 denote the first and second fusion points, respectively. Scale bars = 10 μm. **d** Transmission electron micrographs of the WT, ΔDdSnc1, and DdSnc1[C] ring cells. Black arrows indicate the thickened inner wall of ring cells. Magnified views of the areas denoted using the white dotted box are shown on the bottom or right side. Scale bars = 0.5 μm. These images represent what was observed among at least 5 CRs.

the loss of one resulted in compensatory up-regulation of the other (Fig. 5i, j), further supporting their redundant function. In plants, some homologous SNARE proteins are functionally redundant. For instance, loss of the *A. thaliana SYP121* or *SYP122*, genes encoding closely related SNARE proteins, had little effect on plant growth under diverse conditions, but the growth of the *syp121/syp122* double mutant was significantly reduced[48–50]. However, DdSnc2 is not required for CR inflation (Fig. 4g, h), suggesting that individual Snc proteins also perform gene-specific functions. Sixteen putative SNARE-encoding genes were identified in the genome of *D. dactyloides*[41]. It is likely that other SNARE-encoding genes also participate in PMS formation or other processes associated with CR maturation and inflation (e.g., those discussed below). Exo70, a component of the exocyst complex, is involved in polarized exocytosis and tip growth in fungi[38,51,52]. In the nematode-trapping fungus *Duddingtonia flagrans*, the role of Exo70 in growth and secreting virulence factors has been demonstrated[53]. The deletion of *DdExo70* significantly affected both hyphal growth and CR

morphogenesis in *D. dactyloides*, suggesting its likely role in regulating both the polar growth of mycelium and CR formation.

Although the discovery of PMS and the identification of critical factors/processes required for PMS formation helped improve our understanding of this unique cell morphogenesis process evolved by some carnivorous fungi, the work also presented multiple questions concerning CR maturation and inflation and how other subcellular changes associated with CR inflation occur and are regulated. We have yet to uncover the content of the vesicles forming the PMS and where and how the materials in the vesicles are synthesized and integrate with the PM during ring cell inflation. Besides PM expansion, ring cell inflation also involves cell wall extension and a rapid increase in cell volume. Transverse sections of ring cells show that the cell wall is not uniform in thickness, with the wall being significantly thicker on the inner side (Fig. 2i). After ring cell inflation, the outer cell wall was found dehisced. In contrast, the inner wall appeared stretched but intact (Fig. 1d). The polarized traffic of secretory vesicles provides the

enzymes needed for cell wall construction/remodeling during hypha tip growth (6, 7). When *DdSnc1* was deleted, which blocks multi-vesicle fusions required for PMS formation, the cell wall was still thickened on the inner side of the ring cells (Fig. 6d), suggesting that the vesicles forming the PMS do not contribute to cell wall biosynthesis and thickening. Compared to the outer wall, the stretched inner wall after inflation could be stained well by ConA and GNL, but not CFW (Supplementary Fig. 12), suggesting dextrans rather than chitins may play a crucial role in wall stretching. The elastic guard cell wall, in which the major components are cellulose, xyloglucan, and pectin, is distinct from the cell wall of other cells[54,55]. More detailed analyses of the cell wall structure and composition are needed to uncover the mechanism underpinning the inner wall thickening and wall extension during ring cell maturation and inflation.

Some questions concerning the mechanism underpinning drastic cell volume increase during CR inflation (e.g., signal(s) that trigger CR inflation and which pathway(s) regulate this process) also remain to be explored. Large central vacuoles appeared in inflated CR cells (Fig. 1g), suggesting that vacuole morphogenesis is the primary driver of cell volume increase. Intriguing questions are how the large central vacuole forms so quickly, how its enlargement coordinates with PM expansion, and which signals regulate and drive these rapid morphological changes. Vacuole biogenesis is energetically less costly than generating new cytoplasmic content. Fungal morphogenesis often involves hyphal space-filling using vacuoles[56]. Hyphae of some human fungal pathogens can grow under nutrient-limited conditions by expanding the volume of vacuoles rather than synthesizing new organelles. In *Candida albicans*, germ tube emergence involves a substantial vacuole enlargement in the mother yeast cell and the migration of most of the cytoplasm into the newly formed hypha[56,57]. However, vacuole biogenesis remains poorly understood. The vesicles forming PMS, previously suggested as lysosomes[58], likely supply enzymes and other materials that drive vacuole formation and expansion associated with CR inflation[59].

## Methods

### Strains and growth conditions

The WT *D. dactyloides* strain 29 (CGMCC3.20198) and its mutants were cultured on PDA (BD, NY, USA) and CMA (BD). The nematode *C. elegans* was maintained on Nematode Growth Medium (NGM) covered with *Escherichia coli* OP50 at 23 °C.

### Assays for CR formation and inflation

Conidia harvested from 2-week-old CMA cultures were inoculated on WA (BD) plates. For the mutants that lost the ability to sporulate, ground mycelia were plated[60]. After 3 days of incubation at 25 °C, ~1500 nematodes were evenly distributed in each plate. Immature and mature CRs were obtained after 18 and 72 h, respectively. CRs were also produced using a modified Muller's method[31]. Approximately $5 \times 10^3$ conidia were inoculated in 5 mL 1% horse serum (HyClone, UT, USA) in a 6-cm Petri dish and incubated at 25 °C. Incompletely formed and immature CRs were obtained after 48 h of incubation, while mature CRs were collected after 5 days of incubation.

The application of water at 55 °C or the entry and rubbing of nematodes against the inner lining of the rings can stimulate CR inflation[31]. We used both approaches to measure the inflation of CRs. Conidia were placed on 6-cm WA plates and incubated at 25 °C for 3 days. Approximately 500 nematodes were then evenly distributed on each plate to induce CR formation. After adding water at 55 °C, uninflated and inflated CRs were counted at multiple time points to determine the inflation rate. Alternatively, small round tubes were randomly inserted into the plates to demarcate limited area (~ 0.75 cm²). Subsequently, 2000 nematodes were added to these designated areas, and after a 30-min period, both inflated and uninflated CRs were counted to determine the inflation rate.

### Staining procedures and inhibitor treatments

Conidia were inoculated in 1% horse serum (see above) to prepare incompletely formed, immature, mature, and inflated CRs. The membrane-specific dye FM4-64 (Invitrogen, CA, USA) was dissolved in water to prepare a 0.2 mg/mL stock solution. CR samples placed in a 6 μg/mL FM4-64 solution were incubated for 15 min. CRs were stained using 2 μg/mL CFW (Sigma-Aldrich, MO, USA) for 5 min. Polysaccharides in ring cells were stained using 10 μg/mL lectin ConA (Vector Labs, CA, USA) or 10 μg/mL lectin GNL (Vector Labs) for 30 min. Inflated CRs were treated with neutral red (Solarbio, Beijing, China) for 15 min to visualize vacuoles and check the integrity of ring cells. CRs were incubated in DAPI (Solarbio) for 15 min to observe ring cell nuclei. All stained samples were washed three times with phosphate buffer solution (PBS, pH 7.0) before imaging via a confocal laser scanning microscope (Leica TCSSP5). To examine how BFA (Sigma-Aldrich) affects ring cell inflation, approximately $10^3$ conidia were plated on a Petri dish (2.2 cm in diameter) containing WA. After incubating at 25 °C for 3 days, ~500 nematodes were evenly added to induce CR formation. Immature and mature CRs were incubated in BFA (30 μM and 50 μM) for 24 h. BFA was dissolved in DMSO to prepare a 50 mg/mL stock solution, and the same volume of DMSO was used for control. After removing BFA and DMSO, water at 55 °C was applied to stimulate CR inflation, and the percentage of inflated CRs was determined. The reagents are listed in Supplementary Table 1.

### SEM and TEM analyses

After incubating conidia of strain 29 and its mutants on WA plates for 3 days, nematodes were added to induce CR formation. Agar plugs (0.5 × 0.5 cm) were cut out, washed, and fixed overnight in 2.5% glutaraldehyde in 0.1 M PBS (pH 7.2) at 4 °C. After washing the plugs with PBS three times, they were dehydrated using ethanol gradients (40–100%) and then dried. The specimens were imaged using a scanning electron microscope (QUANTA 200). For TEM analysis, after incubating conidia plated on CMA plates overlaid with a cellophane membrane for 4 days, ~1500 nematodes per plate were evenly distributed. Immature CRs collected after 18 and 22 h of induction and mature CRs collected after 72 h were fixed in 2.5% glutaraldehyde overnight, dehydrated using increasing concentrations of ethanol (40–100%), and embedded in resin for sectioning. After treating the resulting sections with 2% uranium acetate followed by lead citrate, a transmission electron microscope (Hitachi, HT7700) was used to image them.

### Identification of *D. dactyloides* genes likely involved in membrane fusion

Orthologs of DdSnc1, DdSnc2, and DdExo70 were identified through BLASTP searches of NCBI protein databases[61]. PFAM (http://pfam.xfam.org/) and InterProScan (http://www.ebi.ac.uk/interpro/) were used to predict conserved functional domains[62]. Putative transmembrane domains were predicted using TMHMM (https://services.healthtech.dtu.dk/). Sequence alignments of DdSnc1, DdSnc2, and DdExo70 homologs were visualized using Clustalomega (https://www.ebi.ac.uk/Tools/msa/clustalo/)[63].

### Gene deletion and complementation

Genomic DNA was extracted from 7-day-old *D. dactyloides* culture on PDA overlaid with a cellophane membrane. Targeted gene deletion via homologous recombination was performed using *Agrobacterium tumefaciens*-mediated transformation (ATMT)[37]. Each target gene's 5' and 3' flanking regions (2–3 kb for each region) were amplified using primers 5 F/5 R and 3 F/3 R, respectively. Subsequently, the two fragments were inserted into the upstream and downstream of the hygromycin B or geneticin (G418) resistance gene, respectively, in vector pAg1-H3-HygR or pAg1-H3-NeoR (a modified pAg1-H3-HygR vector conferring resistance to geneticin) to produce gene knockout

constructs. For complementing each mutation, the corresponding gene together with its promoter (~2 kb upstream of the start codon) and terminator (~2 kb downstream of the stop codon) regions were amplified using the primers CF and CR. The fragment was then cloned into pAg1-H3-NeoR for transformation. Approximately $10^6$ conidia of *D. dactyloides*, harvested from 2-week-old CMA cultures, were cocultivated with the positive AGL-1 transformant for 7 days. Subsequently, the cocultivations were overlaid with PDA medium containing either 100 μg/mL of hygromycin B or 200 μg/mL of geneticin for the purpose of selection. Transformants were then transferred to PDA supplemented with 100 μg/mL hygromycin B or 200 μg/mL geneticin. All the mutants were verified by PCR, Southern analysis, and RT-PCR. The primers are shown in Supplementary Table 2.

### Subcellular localization of DdSnc1 and DdSnc2
Primers Snc1-R-UP and Snc1-R-DN were used to amplify a 5217 bp fragment covering the *DdSnc1* ORF (980 bp) and the promoter (2030 bp) and terminator r (2207 bp) regions. After amplifying the 708 bp RFP ORF using primers RFP_UP and RFP_DN, the fragment was fused to the C terminus of the *DdSnc1* ORF. The resulting construct was cloned into binary vector pAg1-H3-NeoR for ATMT, and transformants were selected using geneticin. As a control, the RFP gene under the control of the *DdSnc1* promoter was used. Primers Snc2-G-UP and Snc2-G-DN were used to amplify the *DdSnc2* ORF (853 bp). The GFP coding region (717 bp), amplified using primers GFP_UP and GFP_DN, was fused to the C terminus of the *DdSnc2* ORF. This construct was placed under the *β-tubulin* promoter. The resulting construct was cloned into pAg1-H3-HygR. The primers used all gene manipulations are shown in Supplementary Table 2.

### Mutant characterization
To compare the growth rate between strain 29 and its mutants, culture plugs (6 mm in diameter) were placed on PDA and incubated at 25 °C for 8 days. Their colony diameters were measured every 24 h. The culture plugs were placed on CMA plates and incubated at 25 °C for 10 days to compare conidiation. Cultures were scraped off using a glass spatula and placed into 5 mL sterile water. After filtration through three layers of lens tissue, the number of conidia per $cm^2$ of the colony was calculated. Approximately $10^4$ conidia were placed on WA plates to compare CR formation and inflation. After 3 days of growth at 25 °C, about 1,500 nematodes were added to each plate, and the number of CRs formed was counted at 12, 24, and 36 h. After adding water at 55 °C, uninflated and inflated CRs were counted at multiple time points to determine the inflation rate. Approximately $10^3$ conidia were plated on a Petri dish (3.5 cm in diameter) containing WA to quantify their nematode-capturing capability. After 3 days, 200 nematodes per plate were added, and the number of captured nematodes was recorded at 16, 24, and 36 h.

### RNA preparation and RT-PCR analysis
For RNA extraction, conidia were plated on WA overlaid with a cellophane membrane. The resulting mycelia were scraped off using a medicine spoon. Total RNA was isolated using the Trizol reagent (Invitrogen). Reverse transcription was conducted using All-in-One First-Strand cDNA Synthesis SuperMix (TransGen). Real-time PCR was performed using SYBR PCR mix (Roche), and the data were normalized using *β-tubulin* expression. The expression fold change was calculated using the $2^{-\Delta\Delta ct}$ method.

### Statistics and reproducibility
All experiments were repeated ≥2 times, and the resulting data were presented as the mean ± standard deviation. GraphPad Prism 8 (https://www.graphpad.com/scientific-software/prism/) was used for statistical analyses. Asterisks in the figures indicate the level of significance (*$P < 0.05$, **$P < 0.01$, and *** $P < 0.001$) as determined using the two-tailed unpaired Student t test. All experiments were repeated at least two to three times independently with similar results.

### Reporting summary
Further information on research design is available in the Nature Portfolio Reporting Summary linked to this article.

## Data availability
All data are available in the main text or the supplementary materials. The whole-genome shotgun sequence of strain 29 is available in Gen-Bank (accession # JAGTWJ000000000; BioSample SAMN18837316; BioProject # PRJNA723920). All other relevant data supporting the findings of this study are available on request. Source data are provided with this paper.

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

## Acknowledgements

We thank Dr. Di An for helping acquire and interpret TEM images and Dr. Qilin Yu at Nankai University and Dr. Kirk Czymmek at the Donald Danforth Plant Science Center for valuable suggestions. S.K. acknowledges support from the USDA National Institute of Food & Agriculture and Federal Appropriations (Project PEN4839) and the Chinese Academy of Science President's International Fellowship. This study was supported by grants from the National Natural Science Foundation of China grants 32020103001 and 32230004 (to X.L.) and the Fundamental Research Funds for the Central Universities, Nankai University (to X.L.).

## Author contributions

Y.C., M.X., and X.L. designed the experiments. Y.C. conducted most of the experiments. J.L., D.W., and Y.F. performed some experiments and assisted Y.C. All authors contributed to analyzing data. Y.C., S.K., and X.L. wrote the paper.

## Competing interests

The authors declare no competing interests.
