## [Peer Review File · Nature Communications]

A palisade-shaped membrane reservoir is required for rapid ring cell inflation in *Drechslerella dactyloides*REVIEWER COMMENTS

Reviewer #1 (Remarks to the Author):

The paper by Chen et al. describes the analysis of trap formation and function in *Drechlerella dactyloides*. The paper consists of two parts. In the first part the authors characterize the contractile traps, timing, functioning, morphology. Although some of this part has been shown previously (old papers on the structure, but also the genome paper from 2014), Fig. 1 shows beautiful pictures and is a good introduction to the phenomenon. Fig. 2 is also part of the morphological characterization but using SEM to show the presence of palisade membrane structures (PMS) underneath the cytoplasmic membrane of the inner rim of the trap cells. The second part of the paper aims at understanding the phenomenon of rapid cell inflation by PMS fusion at a more mechanistic level. First, the authors show that PMS formation prior to trap maturation is required for full functioning of the traps. Then, they characterized two v-SNAREs and Exo70 and show that Snc1 is required for membrane fusion and cell inflation. Snc1 and Snc2 appear to have redundant functions in normal hyphal growth.

It is an interesting study, and the work has been performed very well. I had reviewed a previous version of the paper when it was submitted to a different journal and find that the authors addressed my concerns at that time.

minor: Exo70 had been characterised in another nematode-trapping fungus. This could be discussed somewhere. PMID: 34735554.

Reviewer #2 (Remarks to the Author):

This article by Chen et al., entitled "Palisade-shaped membrane reservoir formed via multi-vesicle fusion is required for rapid ring cell inflation in a carnivorous fungus" is written exceptionally well and describes a unique SNARE phenomenon by carnivorous fungus *Drechlerella dactyloides*. SNARE proteins are extremely important for vesicular transportation of cellular contents inside and outside the fungal cells. Snc proteins are important exocytic SNARE proteins which play major role during exocytosis and mediate secretion of fungal effectors. The authors have described the role of Snc1, Snc2 and Exo70 proteins in Palisade-shaped Membrane-building Structure (PMS) formation and ring cell inflation in *Drechlerella dactyloides*.

Overall, the authors have done tremendous work supported by experimental proof, however, I have some suggestions which may help further improve the scope and readability of the manuscript.

Line 57-59: Subsequent fusion is directed by two types of SNARE (soluble N-ethylmaleimide-sensitive factor attachment protein receptor) proteins, v-SNARE and t-SNARE, on the vesicle and PM sides, respectively. Please add the following reference here, "Diverse role of SNARE protein Sec22 in vesicle trafficking, membrane fusion, and autophagy. *Cells*, 8(4), p.337".

Line 58-58: Please mention, t-SNARE as target SNARE which is occupied to or localized at PM.

Line 59-60: Please rewrite this sentence as "These SNARE proteins form a four-helix bundle and pull the two membranes in a close proximity"

Line 64-66: The Snc proteins encoded by filamentous fungi are localized in hyphal tips and subapical compartments and regulate exocytosis in *Aspergillus oryzae* and *Trichoderma reesei*. Please mention the latest citation here "SNARE Protein Snc1 Is Essential for Vesicle Trafficking, Membrane Fusion and Protein Secretion in Fungi. *Cells*, 12(11), p.1547".

Line 75-77: The physical mechanism underpinning the function of such specialized organs and the nature and mode of action of signaling molecules involved have been extensively characterized. Please support your claim with relevant literature citations.

Line 171-177: One possible explanation of BFA treatment maybe that BFA generally affect the initial protein secretion pathway (ER-Golgi transportation) and may have slight or no effect at later protein secretion pathway (Golgi to PM).

Reviewer #3 (Remarks to the Author):

In this manuscript by Chen et al., the authors investigated the mechanism of rapid ring-cell inflation and identified a link between a novel structure, PSM, and the ability for cell to inflate in a nematode-trapping fungus. Overall, I found this work very interesting, and it is also the first study that investigates and reveals mechanism at the cellular level. Although there are still many unanswered questions regarding the mechanism of trap inflation, this work did provide insight into this fascinating biological questions.

Below I have some comments and suggestions that I hope will make the story more complete:

1. The authors mentioned that newly formed CRs cannot be inflated by hot water. They only can be inflated after 36 hrs. It is not clear whether in Fig. 1 I and J, the traps were inflated by hot water? The authors should make this clear in the figure legend.
On the same line, hot water is not a biological cue. What happens if the authors measure the inflation rate with *C. elegans*? Do nematodes also trigger inflating on traps that are formed more than 16 hours? Or actually an "immature CR" can inflate to trap a nematodes. Basically, I suggest the authors do the same experiment as Fig. 1 I, and use *C. elegans*.
2. The authors stated that most animals were caught after 36 hours. Thus, I suggest for fig. 1K, the authors also show an image of a plate with a full lawn of trapped *C. elegans*.
3. In the section where the authors tested several inhibitors (like Fig. 3d, and Supplementary Fig. 3 a and c), the percentage of inflated immature CRs were 100% without the drug treatment. This contradict to authors earlier conclusion that "immature CR" lacks PSM and cannot inflate. I found these results very puzzling, and if the "immature CR" that lacks PSM can inflate, then the conclusion of PSM is required for rapid ring-inflation is not correct.
4. Is PSM a membranous structure? Can authors use membrane labeling to examine the PSM structure and check it's colocalization with the Snc1 and Snc2 RFP and GFP fusion proteins? Since these two components are required for the formation of PSM, I am wondering whether Snc1 and Snc2 colocalize with PSM.

Point-by-point response to referees' comments

We would like to extend our heartfelt gratitude to the reviewers for offering insightful suggestions and specific guidance to enhance the quality of our manuscript. Our responses to all the comments and suggestions received are denoted below (highlighted in blue), and all corresponding changes can be found in the revised manuscript (noted using the track change function of MS).

Reviewer #1 (Remarks to the Author):

The paper by Chen et al. describes the analysis of trap formation and function in *Drechslerella dactyloides*. The paper consists of two parts. In the first part the authors characterize the contractile traps, timing, functioning, morphology. Although some of this part has been shown previously (old papers on the structure, but also the genome paper from 2014), Fig. 1 shows beautiful pictures and is a good introduction to the phenomenon. Fig. 2 is also part of the morphological characterization but using SEM to show the presence of palisade membrane structures (PMS) underneath the cytoplasmic membrane of the inner rim of the trap cells. The second part of the paper aims at understanding the phenomenon of rapid cell inflation by PMS fusion at a more mechanistic level. First, the authors show that PMS formation prior to trap maturation is required for full functioning of the traps. Then, they characterized two v-SNAREs and Exo70 and show that Snc1 is required for membrane fusion and cell inflation. Snc1 and Snc2 appear to have redundant functions in normal hyphal growth.

It is an interesting study, and the work has been performed very well. I had reviewed a previous version of the paper when it was submitted to a different journal and find that the authors addressed my concerns at that time.

R: We sincerely appreciate the positive comments.

minor: Exo70 had been characterized in another nematode-trapping fungus. This could be discussed somewhere. PMID: 34735554.

R: We discussed this study (see lines 310-316 in the revised manuscript):

“Exo70, a component of the exocyst complex, is involved in polarized exocytosis and tip growth in fungi^{1,2,3}. In the nematode-trapping fungus *Duddingtonia flagrans*, the role of Exo70 in growth and secreting virulence factors has been demonstrated⁵³. The deletion of *DdExo70* significantly affected both hyphal growth and CR morphogenesis in *D. dactyloides*, suggesting its likely role in regulating both the polar growth of mycelium and CR formation.”

Reviewer #2 (Remarks to the Author):

This article by Chen et al., entitled “Palisade-shaped membrane reservoir formed via multi-vesicle fusion is required for rapid ring cell inflation in a carnivorous fungus” is written exceptionally well and describes a unique SNARE phenomenon by carnivorous fungus *Drechlerella dactyloides*. SNARE proteins are extremely important for vesicular transportation of cellular contents inside and outside the fungal cells. Snc proteins are important exocytic SNARE proteins which play major role during exocytosis and mediate secretion of fungal effectors. The authors have described the role of Snc1, Snc2 and Exo70 proteins in Palisade-shaped Membrane-building Structure (PMS) formation and ring cell inflation in *Drechlerella dactyloides*.

Overall, the authors have done tremendous work supported by experimental proof, however, I have some suggestions which may help further improve the scope and readability of the manuscript.

R: We are grateful for the reviewer's positive comments and insightful suggestions. We improved the overall quality of the manuscript by incorporating these suggestions as indicated below.

Line 57-59: Subsequent fusion is directed by two types of SNARE (soluble N-ethylmaleimide-sensitive factor attachment protein receptor) proteins, v-SNARE and t-SNARE, on the vesicle and PM sides, respectively. Please add the following reference here, “Diverse role of SNARE protein Sec22 in vesicle trafficking, membrane fusion, and autophagy. *Cells*, 8(4), p.337”.

R: We appreciate the suggestion and cited this reference.

Line 58-58: Please mention, t-SNARE as target SNARE which is occupied to or localized at PM.

R: Thank you for the clarification. We revised the sentence to “Subsequent fusion is directed by two types of SNARE (soluble N-ethylmaleimide-sensitive factor attachment protein receptor) proteins: vesicle or v-SNARE associated with the vesicle membrane and target or t-SNARE localized at the PM.” (see lines 52-55 in the revised manuscript)

Line 59-60: Please rewrite this sentence as “These SNARE proteins form a four-helix bundle and pull the two membranes in a close proximity”

R: Done.

Line 64-66: The Snc proteins encoded by filamentous fungi are localized in hyphal tips and subapical compartments and regulate exocytosis in *Aspergillus oryzae* and *Trichoderma reesei*. Please mention the latest citation here “SNARE Protein Snc1 Is Essential for Vesicle Trafficking, Membrane Fusion and Protein Secretion in Fungi. Cells, 12(11), p.1547”.

R: We cited this reference. Thanks for the suggestion.

Line 75-77: The physical mechanism underpinning the function of such specialized organs and the nature and mode of action of signaling molecules involved have been extensively characterized. Please support your claim with relevant literature citations.

R: The following references were added:

25 Mano, H. & Hasebe, M. Rapid movements in plants. *J. Plant Res.* 134, 3-17, (2021).

26 Forterre, Y. Slow, fast and furious: Understanding the physics of plant movements. *J. Exp. Bot.* 64, 4745-4760, (2013).

These two reviews offer valuable insights into the physical mechanisms underlying rapid multicellular movements, including water transport, poroelasticity, and mechanical instabilities. They also cover the gene products responsible for these

processes, especially those related to ion transport. Additionally, the reviews discuss the molecular systems facilitating quick cell-cell communication and the mechanosensing systems that trigger responses.

Line 171-177: One possible explanation of BFA treatment maybe that BFA generally affect the initial protein secretion pathway (ER-Golgi transportation) and may have slight or no effect at later protein secretion pathway (Golgi to PM).

R: Thanks for this insightful explanation regarding the potential effect of BFA treatment on secretion pathways. We incorporated this explanation in lines 167-169 of the revised manuscript.

Reviewer #3 (Remarks to the Author):

In this manuscript by Chen et al., the authors investigated the mechanism of rapid ring-cell inflation and identified a link between a novel structure, PSM, and the ability for cell to inflate in a nematode-trapping fungus. Overall, I found this work very interesting, and it is also the first study that investigates and reveals mechanism at the cellular level. Although there are still many unanswered questions regarding the mechanism of trap inflation, this work did provide insight into this fascinating biological questions.

R: We sincerely appreciate the reviewer's acknowledgment of the contribution of our work in advancing our understanding of the mechanism underpinning CR inflation.

Below I have some comments and suggestions that I hope will make the story more complete:

1. The authors mentioned that newly formed CRs cannot be inflated by hot water. They only can be inflated after 36 hrs. It is not clear whether in Fig. 1 I and J, the traps were inflated by hot water? The authors should make this clear in the figure legend.

On the same line, hot water is not a biological cue. What happens if the authors measure the inflation rate with *C. elegans*? Do nematodes also trigger inflating on traps that are formed more than 16 hours? Or actually an “immature CR” can inflate to trap a nematodes. Basically, I suggest the authors do the same experiment as Fig. 1 I, and

use *C. elegans*.

R: We greatly appreciate these valuable suggestions. In Fig. 1i and j, the CRs were inflated by hot water (55°C) treatment. We also revised the figure legend accordingly.

As suggested, we conducted an additional experiment to measure the CR inflation rate with nematodes at different time points. Conidia were placed on 6-cm WA plates and incubated at 25°C for 3 days. Approximately 500 nematodes were then evenly distributed on each plate to induce CR formation. The application of 55°C water onto the CRs or the entry and rubbing of nematodes against the lumen of the rings could stimulate the inflation⁵. Nematodes pass through the CRs in a random manner. To ensure nematode entry through each CR, we introduced a sufficient number of nematodes (~2,000) within a limited area (~0.75 cm²) and provided them ample crawling time (30 min) (Supplementary Fig. 1a). After approximately 16 h of induction, a large number of CRs formed. However, while these CRs could attract nematodes into their cavities, they were unable to be stimulated by the nematodes to inflate (Supplementary Fig. 1b, c). Additionally, Fig. 1i illustrates the process of nematodes entering and escaping an immature CR at 16 h. With prolonged induction time, the percentage of inflated CRs continuously increased (Supplementary Fig. 1b, c). These results indicated that immature CRs cannot be triggered to inflate by hot water or a passing nematode.

Supplementary Fig. 1. Measurement of CR inflation using nematodes. **a** A diagram illustrating how we measured CR inflation by applying ~2,000 nematodes in each area (noted by blue circles) to ensure nematode entry through all CRs in these areas. **b** Boxplot showing the percentages of inflated CRs after applying nematodes (n = 10). **c** Light micrographs showing CR inflation after applying nematodes. The lower panel shows magnified views of the areas noted by the dotted black box in the upper panel. Red and blue arrows denote uninflated and inflated CRs, respectively. Scale bars = 50 μm .

2. The authors stated that most animals were caught after 36 hours. Thus, I suggest for fig. 1K, the authors also show an image of a plate with a full lawn of trapped *C. elegans*.

R: We incorporated an image showing a substantial number of trapped nematodes in Fig. 1k.

Fig. 1k. Bar chart showing the percentages of nematodes captured at 16, 24 and 36 hours after its introduction ($n = 5$). Data are means \pm SD. Light micrograph showing trapped nematodes after 36 hours. Black arrows denote inflated CRs. Scale bar = 50 μ m.

3. In the section where the authors tested several inhibitors (like Fig. 3d, and Supplementary Fig. 3 a and c), the percentage of inflated immature CRs were 100% without the drug treatment. This contradict to authors earlier conclusion that “immature CR” lacks PSM and cannot inflate. I found these results very puzzling, and if the “immature CR” that lacks PSM can inflate, then the conclusion of PSM is required for rapid ring-inflation is not correct.

R: This confusion was caused by the complex treatments we employed. We exposed immature and mature CRs with inhibitors (Brefeldin A, benomyl and latrunculin B) for 24 h and measured the percentage of inflated CRs after stimulating CR inflation using hot water. In the absence of any inhibitor treatment (0 μ g/mL), immature CRs matured normally during this 24 h period and were able to inflate, reaching a 100% inflation rate in response to hot water treatment. When immature CRs were treated with 30 or 50 μ g/mL BFA for 24 h, there was a decrease in the percentage of inflated CRs (Fig. 3d). This decrease can be attributed to BFA's impairment of vesicle trafficking during maturation stage, potentially affecting the initial trafficking pathway such as ER-Golgi transportation as mentioned by Reviewer #2. Moreover, the treatment did not affect the inflation capacity of mature CRs, as they had already formed the PMS. These results indicated the involvement of vesicular trafficking in CR maturation and that PMS

formation is a prerequisite for ring cell inflation.

4. Is PSM a membranous structure? Can authors use membrane labeling to examine the PSM structure and check its colocalization with the Snc1 and Snc2 RFP and GFP fusion proteins? Since these two components are required for the formation of PSM, I am wondering whether Snc1 and Snc2 colocalize with PSM.

R: The PMS was formed through the fusion of vesicles during CR maturation. Since these vesicles are coated with the membrane, we think that the PMS resembles buds, wrinkles, folds, and invaginations that support membrane expansion in other types of cells ^{6,7}. However, the nature and role of electron-dense materials within the vesicles that form the PMS remain to be determined.

We stained the CRs with the fluorescent membrane dye FM4-64 ⁸ and found that the inner side of ring cells of mature CRs was distinct from the outer side, with more membranes present on the inner side (Supplementary Fig. 3a), supporting that PMS formation brought extra membrane-building materials needed to accommodate the PM expansion caused by CR inflation.

Deletion of *DdSnc2* did not impair CR inflation capacity (Fig. 4g, h), and DdSnc2 did not accumulate on the inner rim of the ring cells (Fig. 5g), where PMS forms, indicating that DdSnc2 is not required for PMS formation. However, it was observed that *DdSnc1* is required for PMS formation. Although simultaneous imaging of PMS and DdSnc1 was not performed, the ultrastructure analysis demonstrated the presence of PMS on the inner side of ring cells (Fig. 2h-j), while DdSnc1 was localized at the same inner side of the ring cells (Fig. 4c). These results collectively support the notion that Snc1 colocalizes with PMS.

References

- 1 Chen, X., Ebbole, D. J. & Wang, Z. The exocyst complex: delivery hub for morphogenesis and pathogenesis in filamentous fungi. *Curr. Opin. Plant Biol.* **28**, 48-54 (2015).
- 2 Riquelme, M. et al. The *Neurospora crassa* exocyst complex tethers

- Spitzenkorper vesicles to the apical plasma membrane during polarized growth. *Mol. Biol. Cell.* **25**, 1312-1326 (2014).
- 3 Giraldo, M. C. et al. Two distinct secretion systems facilitate tissue invasion by the rice blast fungus *Magnaporthe oryzae*. *Nat. Commun.* **4**, 1996 (2013).
- 4 Wernet, N., Wernet, V. & Fischer, R. The small-secreted cysteine-rich protein CyrA is a virulence factor participating in the attack of *Caenorhabditis elegans* by *Duddingtonia flagrans*. *PLoS Pathog.* **17**, e1010028 (2021).
- 5 Muller, H. G. The constricting ring mechanism of two predacious hyphomycetes. *Trans. Br. Mycol. Soc.* **41**, 341-364 (1958).
- 6 Sinha, B. et al. Cells respond to mechanical stress by rapid disassembly of caveolae. *Cell* **144**, 402-413 (2011).
- 7 Groulx, N., Boudreault, F., Orlov, S. N. & Grygorczyk, R. Membrane reserves and hypotonic cell swelling. *J. Membr. Biol.* **214**, 43-56 (2006).
- 8 Liu, J., Tong, S. M., Qiu, L., Ying, S. H. & Feng, M. G. Two histidine kinases can sense different stress cues for activation of the MAPK Hog1 in a fungal insect pathogen. *Environ. Microbiol.* **19**, 4091-4102 (2017).

REVIEWERS' COMMENTS

Reviewer #3 (Remarks to the Author):

The authors have addressed my questions and comments sufficiently. I now fully support the manuscript to be accepted.

Point-by-point response to referees' comments

Reviewer #3 (Remarks to the Author):

The authors have addressed my questions and comments sufficiently. I now fully support the manuscript to be accepted.

R: We sincerely appreciate your feedback and are glad to hear that our responses have sufficiently addressed your questions and comments.